# Factors associated with quality of life (QOL) scores among methadone patients in Myanmar

**Sun Tun** [1,2] *, **Vicknasingam Balasingam** [2], **Darshan Singh Singh** [2]

**1** Myanmar Medical Association, Yangon, Myanmar, **2** Centre for Drug Research, Universiti Sains Malaysia, Penang, Malaysia

* suntun1@gmail.com

## Abstract

The Drug Dependency Treatment and Research Unit (DDTRU) in Myanmar established opioid substitution with methadone in 2006. Reducing HIV transmission could be affected by eliminating the unsafe needle sharing among injecting drug uses and treatment with opioid substitution. The quality of life (QOL) among the clients retained in the methadone program is important for their personal development and is an indication of the treatment efficacy. This study evaluated factors associated with the QOL of methadone patients to ensure efficient service delivery. It also identified how patients' characteristics had differed QOL scores of respondents. This cross-sectional study was conducted in five cities with stratified random sampling. The study assessed the QOL of methadone patients in Myanmar. The study recruited 210 respondents to answer structured questionnaires for their quality of life: WHO-QOL-BREF questionnaires and urine sample collection for methadone and illicit drug use. Survey responses on the QOL were transformed into 100-scale ratings, and higher QOL scores reflect better QOL. The average score of total QOL was 60.82%; precisely 60.09% in the physical domain, 63.11% in the psychological domain, 59.87% in the social relation domain, 60.41% in the environmental domain respectively. Respondents who reported illicit drug use had lower QOL scores. Statistically significant association of the QOL category of the methadone patients was identified with frequent methadone treatment episodes, the infection status of HIV, current treatment on antiretroviral therapy (ART), tuberculosis (TB) treatment history, sexually transmitted infections (STI) history in their lifetime, current work status as peer, Addiction Severity Index (ASI) for drug use, satisfaction with current marital status, satisfaction with current leisure status, history of psychological abuse within 30 days, heroin injection within 30 days, frequency of injection, and reported use of barbiturates (p<0.05). Addressing these factors will improve the treatment service intervention and the quality of life among methadone patients.

## Introduction

There were an estimated 62 million opioid users worldwide in 2019 [1]. According to the World Drug Report 2021, Myanmar still accounts for 14% of the world's total opium

---

**Data Availability Statement:** The data used to support the findings of this study is available at 10.6084/m9.figshare.19663050.

**Funding:** The authors received no specific funding for this work.

**Competing interests:** The authors have declared that no competing interests exist.

production. Out of 93,000 estimated people who inject drugs in Myanmar, 24,666 PWID were taking methadone for Opioid substitution in June 2021 [2]. Compared to healthy adults, opioid users had a significantly poor quality of life (QOL), especially in the physical, psychological, and social domains (P<0.001) [3]. WHO defined "Quality of Life" as an individual's perception of their position in life in the context of the culture and value systems in which they live and in relation to their goals, expectations, standards and concerns [4]. Quality of Life is a multidimensional concept which incorporates the individual's perception of themselves and other aspect of life. WHO developed a quality of life assessment, WHOQOL for a genuinely international measure of quality of life for continued promotion of a holistic approach to health and health care [5].

As drug dependence is a multi-factorial health disorder with relapsing and remitting in nature, comprehensive intervention packages were also addressed in Myanmar. The methadone programme started and enrolled 260 patients in Myanmar in 2006, according to the report of the Drug Dependency Treatment and Research Unit [6]. Myanmar had 55 methadone sites (including 15 one-stop-shop sites) [7]. Methadone treatment was effective and associated with less opioid use [8] and improved the quality of life of the patients at six months after treatment [9] as well as 12 months follow up [10]. The Program supports improving QOL among methadone patients, and it is also necessary to understand the factors affecting QOL among methadone patients in Myanmar. As public and private sectors have been supporting harm reduction work with substitution therapy in recent years, this study evaluates and identifies the factors affecting the patients' quality of life. This evaluation is necessary for developing the intervention in an improved situation. The Covid-19 pandemic in 2020 and political changes in 2021 interrupted the ongoing service expansion and restricted the momentum of service delivery of methadone provision. There was a reduction of clients taking methadone in 2021 compared to 2020 data.

To assess the general QOL relating to the patient satisfaction of the drug users, WHOQOL-100 and WHOQOL-BREF were mostly used in the studies of patients treated for substance abuse and addiction [11]. The use of WHOQOL-BREF, a short-form version of the WHO-QOL-100 quality of life assessment, covered an individual's overall perception of quality of life in a conceptual coherence and health, as well as four domain scores on the person's physical health, psychological health, social relationships and their relationship to salient features of the environment [5]. The WHOQOL-BREF was the most commonly used test for patients treated with methadone for its reliability and validity [12]. Patients with opioid use disorder had a poor QOL due to their suffering from other chronic diseases [9]. However, those patients with a better QOL were associated with successful methadone treatment outcomes [13].

This study identified the associated characteristics of the respondents with the quality of life of the methadone clients. It also determined how these characteristics had differed the QOL scores. This analysis was the first of such kind of WHOQOL-BREF study for methadone patients in Myanmar. Those patients who took methadone at least 6 months in the methadone program were defined as retained clients by the DDTRU. As the patients in the MMT (Methadone Maintenance Treatment) were taking the methadone for a minimum of 6 months, this study aimed to identify the reported QOL and how QOL differed from the reported characteristics of methadone treatment by having at least a considerable duration.

## Methods

### Study design, respondents and location

This study determines the factors and parameters associated with patients' quality of life enrolled in methadone treatment. The study was a cross-sectional study with 210 respondents recruited from five cities in Myanmar (Yangon, Mandalay, Lashio, Kawlin and Mohnyin). A

sample of 42 persons was recruited from each site with a minimum of 6-month duration on methadone. An equal number of respondents was recruited from selected MMT sites as a quota sampling technique until the fulfilment of the sample size. From the eligible candidate list of the service providers/ resource people of the drop-in-centres, the respondents were recruited randomly and explained to them the objectives and benefits of the study. As the interested community was vulnerable and marginalised, respondents were recruited for participation who gave consent to participate in the survey interview and those who agreed to provide urine specimens linked with their response. Individual interview process and urine sample collection were done at the drop-in-centres or private rooms of the restaurants for the privacy of the assessment. The study recruited 210 respondents to answer personal interviews with 26-item survey questionnaires relating to their quality of life: WHOQOL-BREF questionnaires and urine sample collection for methadone and identifying illicit drug use.

## Inclusion and exclusion criteria

Patients above 18 years old who consented to participate in the study were verified with the methadone test kit and confirmed their participation. The inclusion criteria specified those respondents who were at least 6-month treatment continuously in the recent enrolment on methadone treatment. However, there was no restriction on the first time of methadone treatment or additional episodes of the methadone treatment. Further urine tests of the drugs (Morphine, Cannabis, Methamphetamine, Amphetamine, and Benzodiazepine) were proceeded. We then collected physical measures and surveyed questionnaires to collect the demographic information, previous drug use history, and reported infectious disease status and methadone treatment profile. Those patients who were not fit enough to respond to the survey with psychological and medical conditions were excluded. However, we did not exclude anyone with psychological or medical conditions during the interview process.

**Data collection.**   Data collection was done in five selected cities from May 2017 to July 2017. These five cities were from five States and Regions where methadone dispensing sites existed in the country. Myanmar had 55 methadone sites (including 15 one-stop-shop sites). Among these sites, sites selection was done with stratified random sampling from eligible cities, of States and Regions [7].

**Instrument development and measures.**   This study used a semi-structured questionnaire and validated instruments to evaluate the respondent's quality of life and well beings of the patients. The study questionnaire was comprised of semi-structured survey questionnaires developed from validated questionnaires of 1: WHO QOL BREF [14] which included Physical Health, Psychological, Social Relationships and Environmental domains; 2: Timeline Follow Back (TLFB) [15], which was used to determine respondents drug use frequency in the last seven days, and 3: Addiction Severity Index-Lite (ASI) [16] included composite scores for employment, alcohol, drug use, legal status, family/social status (while assessment did not include medical status and psychiatric status).

The WHOQOL-BREF instrument: an abbreviated version of the WHOQOL-100, which covered 26 questions in four domains: physical, psychological, social relationships and environment. Responses to these 26 questions were recorded on 5-point Likert scale and transformed each raw scale score to a 0–100 scale using the formula.

These are the steps for checking and cleaning data and computing domain scores for the WHOQOL-BREF.

Step 1: Check all 26 items from the assessment having a range of 1–5

Step 2: Convert 3 negatively phrased items into relevant positively framed questions.

Step 3: Compute domain scores from relevant domain questions

Step 4: Transform scores to a 0–100 scale

This step involves transforming each raw scale score to a 0–100 scale using the formula shown below:

$$Transformed\ Scale = \left[ \frac{(Actual\ raw\ score - lowest\ possible\ raw\ score\ )}{Possible\ raw\ score\ range} \right] \times 100$$

where "Actual raw score" is the value achieved through summation, and "lowest possible raw score" is the lowest possible value that could occur through summation. "Possible raw score range" is the difference between the maximum possible raw score and the lowest possible raw score [17]. The responses were categorised and analysed into four QOL domains, average and total scores. These calculated QOL scores according to different domains and in-total scores were analysed with response characteristics to identify the association with QOL scores.

The validated questionnaires were translated from English to Burmese by two trained bilingual translators and translated back from Burmese to English to reflect the proper understanding and meaning of the original questions. Translated questionnaires were re-tested and noted a clear understanding of the responses. As the limited reviewers validated the translation of questionnaires, the Cronbach's Alpha coefficient was not calculated. There was no statistical validation to compare the comparability of language and similarity of interpretability.

**Urinalysis.** This study included identifying methadone and common illicit drugs in the urine to verify biological parameters. A urine-drug screen was done for methadone, morphine (opioid), cannabis (THC), methamphetamine, amphetamine and benzodiazepine. Linked anonymous testing was applied to ensure the confidentiality of the patient urine sample with their responses. Urine samples were disposed properly once the researchers had recorded the results.

## Statistical analysis

Data analysis was done using Stata 14.0 software. First, we did descriptive statistics for the result of the respondents' QOL. The Chi-square test was used for identifying associations with interested variables. Independent sample t-tests were used for examining the differences between mean scores of the interesting parameter (total QOL scores). We did multiple logistic regression to identify the predictors of the outcome "QOL total score" and stepwise regression analysis at p = 0.05 to recheck the regression output. Statistical significance was set at p<0.05 with a two-tailed analysis. The analysis process omitted the incomplete entries in the calculation and missing data were not counted in the formula calculation to have a reliable estimate and reflected the actual QOL status of the respondents.

## Ethical measures

Proper arrangements for the survey interview and urine sample collections were applied to ensure the privacy and confidentiality of the respondents. Ethical approval was granted by the Human Ethics Committee of the Universiti Sains Malaysia (University of Science, Malaysia) (No. USM/ JEPeM/16080269) and the Department of Medical Research, Myanmar Ministry of Health and Sports (No: Ethics/DMR/2017/057) [18]. As the participants were the most-at-risk population and vulnerable, the research team requested a waiver of the local ethical review committee from participant signing, and it was granted for verbal consent procedure. The interviewer read the informed consent form for the participant and noted the obtained individual verbal consent. No particular identification was collected. All survey data and biological samples were analysed anonymously.

Respondents' personal identifiable information was not elicited to mitigate the participation risk. Respondents were informed that all collected urine specimens would be immediately disposed of and their urine drug results would not be disclosed to anyone. The survey procedure was conducted in private rooms of the harm reduction drop-in-centres and private rooms of the restaurant where the respondents did not feel any external threat and to ensure the privacy of the interviews and urine-drug screen. To avoid the legal consequences, we discarded all the urine specimens properly at the end of the interviews. We also compensated respondents for their participation in the study.

## Results

### Demographic characteristics

A total of 210 respondents who reported on methadone maintenance treatment for at least six months were recruited, tested for methadone and analysed. There were 207 males (98.57%) and three females (1.43%). Among the respondents, 209 took methadone daily from the methadone dispensing sites, and 1 (0.48%) had a take-home dose and all had a history of opioid abuse. Respondents had an average age of 33 years ranging from 20 to 76 years. Their average BMI was 20.5, ranging from 14.0 to 33.3. Only 6 (2.86%) had non-formal education, while 159 (75.57%) had primary through high school education, while 45 (21.43%) had college level. With regards to marital status, 84 (40.58%) were married, 27 (13.04%) were separated/divorced, and 96 (46.38%) were single. Employment status was asked; 192 (93.43%) had recent jobs (previous 3-year period), and 18 (8.57%) were in no-job categories (including disabled, students). Twenty-eight (13.46%) had been involved with the income from the drug negotiation within 30 days. Twenty-nine (13.81%) reported their work as outreach workers or peer-educators for drug users. The majority, 173 (83.17%), were on their first-time methadone treatment. The average methadone dose of the patients was 83mg ranging from 20mg to 300mg with an average duration of 28 months (range:6–127 months).

### Reported infection status of the patients

HIV was reported from 74 patients (37%), and 34 (16.27%) had co-infection of HIV and Hepatitis C. Hepatitis C alone was reported by 71 (47.97%), and Hepatitis B was reported by 15 (8.29%). Forty-five (21.53%) reported sexually transmitted infections in their lifetime.

Out of HIV reactive cases, 68 (91.89%) were on antiretroviral therapy (ART), and the average ART duration was 30 months ranging from 1 to 132 months. A high methadone dose of more than 80mg was associated with antiretroviral therapy (p = 0.039).

### WHOQOL scores of methadone patients

The patients' responses on their quality of life were rated with the 1–5 Likert scale mentioned in Table 1. If the average rating was higher than 3 in all 26 QOL questionnaires, the respondent's total response average would be above 79. However, there can be far higher or lower responses than the middle rating of 3. In this analysis, the total of the average rating higher than 79 was calculated as a higher QOL total score. The higher WHO-QOL scores represent better quality of life and lower scores reflect poor QOL.

### Factors associated with the quality of life (QOL) total score

The relationship between respondents' QOL and characteristics of methadone treatment was also evaluated. Based on the interested parameters of the respondents, the Chi-square test for an association of the parameter and QOL total score (Categorised by satisfactory rating; 79

**Table 1. WHO-BREF QOL scores of the respondents.**

| Variable | Frequency (n and %) | Mean (SD) |
|---|---|---|
| Physical Domain (TRANSFORMED to 100 scale) | 209 (100%) | 60.09 (12.85), Range (7–86%) |
| Psychological Domain (TRANSFORMED to 100 scale) | 209 (100%) | 63.11 (15.40), Range (8–100%) |
| Social Relations Domain (TRANSFORMED to 100 scale) | 209 (100%) | 59.87 (18.75), Range (0–100%) |
| Environment Domain (TRANSFORMED to 100 scale) | 209 (100%) | 60.41(13.46), Range (3–94%) |
| All 4 domains average QOL | 209 (100%) | 60.82 (12.16), Range (7–88%) |
| WHO Overall QOL scores | 209 (100%) | 89.41 (12.52), Range (34–119) |
| Q1. Quality of life | 207 (95.69%) | 3.35(1.02) |
| Q2. Satisfaction of health | 209 (100%) | 3.63 (0.97) |
| Q3. Pain | 209 (100%) | 3.03 (1.18) |
| Q4. Medical dependency | 209 (100%) | 3.68 (0.98) |
| Q5. Enjoyment of life | 208 (99.52%) | 3.65 (0.89) |
| Q6. Life meaningfulness | 209 (100%) | 3.30 (0.95) |
| Q7. Concentration | 208 (99.52%) | 3.50 (0.97) |
| Q8. Felt secure | 209 (100%) | 3.77 (0.98) |
| Q9. Physical environment | 209 (100%) | 3.47 (1.00) |
| Q10. Enough energy | 209 (100%) | 3.57 (0.98) |
| Q11. Bodily appearance | 208 (99.52%) | 3.69 (0.97) |
| Q12. Money to meet need | 209 (100%) | 2.62 (0.79) |
| Q13. Availability of information | 207 (95.69%) | 2.95 (0.94) |
| Q14. Leisure activities | 209 (100%) | 3.24 (1.11) |
| Q15. Able to get around | 209 (100%) | 3.88 (0.82) |
| Q16. Satisfaction of sleep | 209 (100%) | 3.81 (0.83) |
| Q17. Satisfaction of daily activities | 209 (100%) | 3.72 (0.76) |
| Q18. Capacity for work | 208 (99.52%) | 3.51 (0.93) |
| Q19. Self-esteem | 208 (99.52%) | 3.40 (0.99) |
| Q20. Personal relationships | 209 (100%) | 3.69 (0.97) |
| Q21. Sex life satisfaction | 205 (97.61%) | 3.49 (0.97) |
| Q22. Social support | 206 (98.56%) | 3.12 (1.07) |
| Q23. Living place/ home | 208 (99.52%) | 3.77 (0.90) |
| Q24. Access to healthcare | 208 (99.52%) | 3.95 (0.74) |
| Q25. Transport | 209 (100%) | 3.61 (0.85) |
| Q26. Negative feelings | 208 (99.52%) | 2.36 (0.77) |

and above) was calculated. There was a significant association between the QOL category of the methadone patients with a significant number of the characteristics namely; frequency of methadone treatment, the infection status of HIV, current treatment on ART, TB treatment history, STI treatment history, current work status as peer/ ORWs, Addiction Severity Index (ASI) for drug use, satisfaction with current marital status, satisfaction with current leisure status, history of psychological abuse within 30 days, heroin injection within 30 days, frequency of injection, and reported use of barbiturates (p<0.05). This showed that those respondents who suffered from infectious diseases related to drug use and blood-borne infections related to injection, who were still on drug use and affecting the psychological response without having the satisfied leisure statuses had significantly associated with QOL issue. The data was further analysed t-test for two independent samples for the differences in the quality of life (QOL) total score against the respondents' characteristics in Table 2. Then, these findings showed clearly which characteristics of the respondents made higher or lower QOL total scores.

**Table 2. Table showing the quality of life (QOL) total score differences by the respondents' characteristics.**

| Respondent characters | Sub groups | Number (n) | Total QOL score (SD) | (p value) |
|---|---|---|---|---|
| Current methadone dose categories | less than or equal 80mg<br>more than 80mg | 132<br>76 | 89.72 (11.52)<br>88.88 (14.23) | 0.6439 |
| Methadone duration | less than or equal 2.4 years<br>more than 2.4 years | 120<br>89 | 89.13 (12.74)<br>89.79 (12.28) | 0.7066 |
| HIV status | Not infected<br>Infected | 126<br>74 | 91.47 (10.76)<br>86.65 (14.54) | 0.0080** |
| Hepatitis C status | Not infected<br>Infected | 77<br>71 | 92.52 (13.14)<br>87 (12.88) | 0.0109** |
| Hepatitis B status | Not infected<br>Infected | 166<br>15 | 89.77 (12.84)<br>89.60 (11.95) | 0.9604 |
| TB treatment history | Not treated<br>treated | 147<br>54 | 90.73 (12.31)<br>86.00 (13.05) | 0.0184** |
| STI history | Not infected<br>Infected | 164<br>45 | 90.15 (12.01)<br>86.69 (14.04) | 0.1003 |
| Age | Younger and equal 35 years<br>Older than 35 years | 128<br>81 | 90.94 (12.82)<br>86.99 (11.70) | 0.0259** |
| BMI (Body Mass Index) | Less than mean BMI (20.5)<br>More than mean BMI | 122<br>84 | 88.32 (13.30)<br>91.45 (11.00) | 0.0766 |
| Currently on ART (Anti-retrovirals) | No<br>On treatment | 141<br>68 | 90.87 (10.96)<br>86.38 (14.90) | 0.0149** |
| Education | Up to primary<br>More than primary | 48<br>161 | 90.19 (14.65)<br>89.17 (11.86) | 0.6237 |
| Recent in employment | Unemployed<br>Employed | 24<br>182 | 81.25 (16.41)<br>90.48 (11.59) | 0.0006*** |
| Current work as Peer/ Outreach | No<br>Peer/outreach | 180<br>29 | 89.73 (12.40)<br>87.38 (13.31) | 0.3487 |
| ASI for Employment | Low score<br>High score | 102<br>104 | 89.92 (12.15)<br>88.70 (13.02) | 0.4880 |
| ASI for Alcohol Use | Low score<br>High score | 28<br>36 | 93.18 (11.10)<br>90.64 (13.56) | 0.4249 |
| ASI for Drug Use | Low score<br>High score | 162<br>47 | 90.88 (11.57)<br>84.32 (14.37) | 0.0014** |
| ASI for Legal Status | Low score<br>High score | 14<br>14 | 91.07 (14.25)<br>84.42 (6.95) | 0.1290 |
| ASI for Family/ Social Status | Low score<br>High score | 139<br>70 | 90.37 (11.38)<br>87.50 (14.41) | 0.1185 |
| Marital status | Currently married<br>Single/separated | 84<br>122 | 90.25 (11.10)<br>88.93 (13.54) | 0.4626 |
| Income | Lower (than average)<br>Higher | 131<br>77 | 88.27 (13.10)<br>91.27 (11.38) | 0.0962 |
| Current marital status satisfaction | Not satisfied<br>Satisfied | 20<br>189 | 80.00 (19.08)<br>90.40 (11.24) | 0.0004*** |
| **Current leisure status satisfaction** | Not satisfied<br>Satisfied | 29<br>180 | 79.28 (16.37)<br>91.04 (11.00) | 0.0000*** |
| Current leisure status with family | Not satisfied<br>Satisfied | 119<br>90 | 87.29 (13.77)<br>92.21 (10.05) | 0.0046** |
| Current leisure status with friend | Not satisfied<br>Satisfied | 135<br>74 | 89.05 (12.01)<br>90.05 (13.46) | 0.5812 |
| Current leisure status alone | Not satisfied<br>Satisfied | 159<br>50 | 91.18 (11.80)<br>83.78 (13.20) | 0.0002*** |
| **Abuse within 30 days** | | | | |
| Psychological abuse | Not experienced<br>Experienced | 183<br>22 | 89.91 (11.82)<br>84.50 (17.15) | 0.0559 |

*(Continued)*

**Table 2.** (Continued)

| Respondent characters | Sub groups | Number (n) | Total QOL score (SD) | (p value) |
|---|---|---|---|---|
| Physical abuse | Not experienced<br>Experienced | 203<br>1 | 89.29 (12.60)<br>89.00 (-) | - |
| Sexual abuse | Not experienced<br>Experienced | 204<br>1 | 89.30 (12.58)<br>96.00 (-) | - |
| **Urine illicit drug findings** (Absent vs Present) | | | | |
| Urine Morphine | Absent<br>Present | 93<br>116 | 91.06 (10.56)<br>88.08 (13.80) | 0.0866 |
| Urine THC | Absent<br>Present | 185<br>24 | 89.44 (12.84)<br>89.13 (9.92) | 0.9071 |
| Urine Methamphetamine | Absent<br>Present | 158<br>51 | 89.70 (12.91)<br>88.51 (11.32) | 0.5576 |
| Urine Amphetamine | Absent<br>Present | 191<br>18 | 89.54 (12.69)<br>87.94 (10.80) | 0.6054 |
| Urine Benzodiazepine | Absent<br>Present | 137<br>72 | 90.57 (11.99)<br>87.19 (13.28) | 0.0639 |
| Last injection within 30 days | No<br>Yes | 93<br>115 | 92.33 (11.38)<br>86.94 (12.93) | 0.0018** |
| Needle sharing within 30 days | Not shared<br>Shared | 99<br>4 | 89.70 (14.85)<br>91.75 (2.63) | 0.7839 |
| Life time sharing of needle and syringes | Not shared<br>Shared | 106<br>102 | 90.46 (13.20)<br>88.25 (11.78) | 0.2053 |
| **Reported drug use status (7 days)** | | | | |
| Drug and alcohol | Not used<br>Used | 47<br>161 | 89.93 (14.19)<br>89.18 (12.04) | 0.7167 |
| Alcohol | Not used<br>Used | 143<br>66 | 88.33 (12.79)<br>91.74 (11.68) | 0.0668 |
| Heroin | Not used<br>Used | 117<br>92 | 90.23 (12.52)<br>88.36 (12.51) | 0.2843 |
| Methadone (210) | Used | - | - | - |
| Morphine | Not used<br>Used | 204<br>5 | 89.57 (12.58)<br>82.60 (7.44) | 0.2194 |
| Benzodiazepine | Not used<br>Used | 186<br>22 | 89.88 (12.48)<br>85.55 (12.73) | 0.1262 |
| Barbiturate | Not used<br>Used | 205<br>4 | 89.65 (12.47)<br>76.75 (8.62) | 0.0409** |
| Antidepressant | Not used<br>Used | 205<br>4 | 89.62 (12.52)<br>78.25 (6.08) | 0.0719 |
| Cocaine (210) | Not used | - | - | - |
| Amphetamine | Not used<br>Used | 156<br>53 | 90.61 (12.52)<br>85.87 (11.96) | 0.0169** |
| THC | Not used<br>Used | 186<br>23 | 89.56 (12.75)<br>88.04 (10.63) | 0.5812 |
| Ecstasy | Not used<br>Used | 206<br>3 | 89.42 (12.60)<br>88.67 (4.93) | 0.9182 |
| Inhalants | Not used<br>Used | 133<br>75 | 90.34 (13.00)<br>87.83 (11.60) | 0.1662 |
| More than one drug | Not used<br>Used | 50<br>142 | 89.94 (15.18)<br>88.70 (11.81) | 0.5568 |

Independent t-test p value

** significance <0.05

*** significance <0.001.

### QOL total score differences by different characteristics

**QOL total score differences by infectious status.** HIV infection among methadone patients affected QOL total scores (p = 0.0080), and this also applied to those on Anti-retrovirals (p = 0.0149). Reported hepatitis C and tuberculosis infection also lowered QOL total scores, especially physical and psychological scores. Meanwhile, STI infection significantly lowered the physical QOL score (p = 0.0125) but not the total QOL score (p = 0.1003).

**QOL total score differences by demographic factors.** The age group of more than 35-year-old had lower QOL total scores (p = 0.0259) than the younger group, and lower scores were also seen in physical, psychological and environmental scores. Employment was also an important factor, and unemployment significantly lowered the QOL total scores (p = 0.0006), and subsequent lower scores were noted in all domain scores. Marital status also affected social QOL, and those who were single/ separated had lower scores than currently married individuals (p = 0.0027). Those satisfied with their current marital status (regardless of marriage or not) had a higher score than those not satisfied (p = 0.0004). Those respondents satisfied with their current leisure habits had significantly higher total QOL than those who were not (p<0.001). It was seen among current leisure status with family (0.0046) and patients satisfied with current leisure status alone (p = 0.0002).

**QOL total score differences by Addiction Severity Index.** The higher score in the addiction severity index showed worsened situation of the respondents. Those who had high addiction severity index by drug use affected the QOL total scores (p = 0.0014), and lower scores in all domain scores. Those with a high addiction severity index by family/ social status had lower social domain scores (p = 0.0029). Patients who experienced psychological abuse had a significantly lower score in psychological (p = 0.0469) and social domains (p = 0.0288).

**QOL total score differences by reported drug use status.** Total QOL was high among those who didn't inject heroin within 30 days (p = 0.0018) and noticed significant QOL differences among physical, psychological and environmental domains. Those who shared needles and syringes in their lifetime had significantly lower scores on the physical QOL domain (p = 0.0206), which could be concurrently seen among those with infection history of HIV, Hepatitis C, TB and sexually transmitted infections (STIs). With the patient-reported drug use within seven days, Barbiturate (p = 0.0409) and Amphetamine (p = 0.0169) had lower QOL total scores, while the latter had shown lower scores in physical (p = 0.0345), psychological (p = 0.0306) and environmental domains (p = 0.0367). In addition to those drugs, the physical domain of QOL was affected by morphine (p = 0.0104), benzodiazepine (p = 0.0268) and antidepressants (p = 0.0003).

### Regression analysis

After considering significant associated factors in the model affecting the QOL total score, stepwise regression was done by considering the significant parameters in the model for identifying final predictors to the outcome "QOL total score". This regression model was constructed based on the significantly associated factors with the QOL score. The backward stepwise regression was applied for those variables which were not significant while taken into consideration in the regression analysis.

In Table 3, the result of the binary logistic regression was described, and we analysed the associated characteristics to the effect changes of WHOQOL total scores. In this adjusted model of binary logistic regression analysis, frequent methadone enrollment of the clients had 3 times reduction (p = 0.018), reported history of STI infection in their lifetime had 2.7 times reduction (p = 0.016), and barbiturate use within seven days had 14 times reduction (p = 0.043) respectively in achieving QOL total score. Meanwhile, a one-unit change in

**Table 3. Table showing the association between the QOL total score and significant characteristics among methadone patients.**

| Variable | Frequency (n, %) | WHO QOL total score | | Adjusted OR (95% CI) * | p value |
|---|---|---|---|---|---|
| | | Low (%) 41 (19.62%) | High (%) 168 (80.38%) | | |
| **Frequency of MMT treatment** | 208 (100%) | | | | |
| First time | 173 (83.17%) | 28 (68.29%) | 145 (86.83%) | 1.0 | |
| More than one | 35 (16.83%) | 13 (31.71%) | 22 (13.17%) | 0.33 (0.14–0.83) | 0.018** |
| **STI history in their lifetime** | 209 (100%) | | | | |
| No | 164 (78.47%) | 26 (63.41%) | 138 (82.14%) | 1.0 | |
| Yes | 45 (21.53%) | 15 (36.59%) | 30 (17.86%) | 0.36 (0.15–0.83) | 0.016** |
| **Reported Barbiturate use** | 209 (100%) | | | | |
| No | 205 (98.09%) | 38 (92.68%) | 167 (99.40%) | 1.0 | |
| Yes | 4 (1.91%) | 3 (7.32%) | 1 (0.60%) | 0.07 (0.01–0.92) | 0.043** |
| **Current leisure status satisfaction** | 209 (100%) | | | | |
| Not satisfied | 29 (13.88%) | 13 (31.71%) | 16 (9.52%) | 1.0 | |
| Satisfied | 180 (86.12%) | 28 (68.29%) | 152 (90.48%) | 3.68 (1.42–9.53) | 0.007** |

Note: OR = odds ratio; CI = confidence interval

** p<0.05

*** p<0.001.

* All ORs were adjusted for frequency of methadone treatment, the infection status of HIV, current treatment on ART, TB treatment history, STI history in their lifetime, current work status as peer/ ORWs, Addiction Severity Index (ASI) for drug use, satisfaction with current marital status, satisfaction with current leisure status, history of psychological abuse within 30 days, heroin injection within 30 days, frequency of injection, and reported use of barbiturates.

satisfaction with current leisure status contributed 3.68 times increase for getting a high QOL total score (p = 0.007). To predict the client characteristics impacted on the total score of QOL, retention variables in stepwise binary logistic regression were analysed. Checking for multicollinearity was done, and the mean-variance inflation factor (vif) was 3.02, and no variable had more than 10. In this stepwise regression analysis, alpha ratio was set at 0.05.

After adjusting for potential confounding variables, the model estimated the association of independent variables; frequent methadone enrollment of the clients, history of STI infection and barbiturate use within seven days were found out to be addressed for the outcome variable of improving the QOL score of the methadone patients. Additionally, current leisure status satisfaction contributed to a higher quality of life of the methadone patients, as shown in Fig 1.

## Discussion

This study elaborated on factors affecting the quality of life among methadone patients and discussed the study findings with other evidence. There were significant QOL total score differences between respondents with infection and non-infection groups. Patients infected with HIV affected their QOL improvement, mainly in psychological and social relationships than physical domain [9]. This cross-sectional study showed no significant difference between different methadone duration and methadone dose groups to QOL total score (p>0.05). However, cohort findings proved an improvement in QOL score with the higher dose of methadone and longer duration of treatment [13]. Regarding the physical measurement of the respondents, Body Mass Index (BMI) was not associated with the QOL total score (p = 0.0766). However, it was reported that less BMI group was associated with physical, psychological and social domains of QOL [19]. Among our respondents, 116 (55.5%) admitted that they had injected heroin in the last 30 days, and it was also seen as 45.6% reported injections in the past month [20]. Current analysis proved that those who didn't inject heroin in the

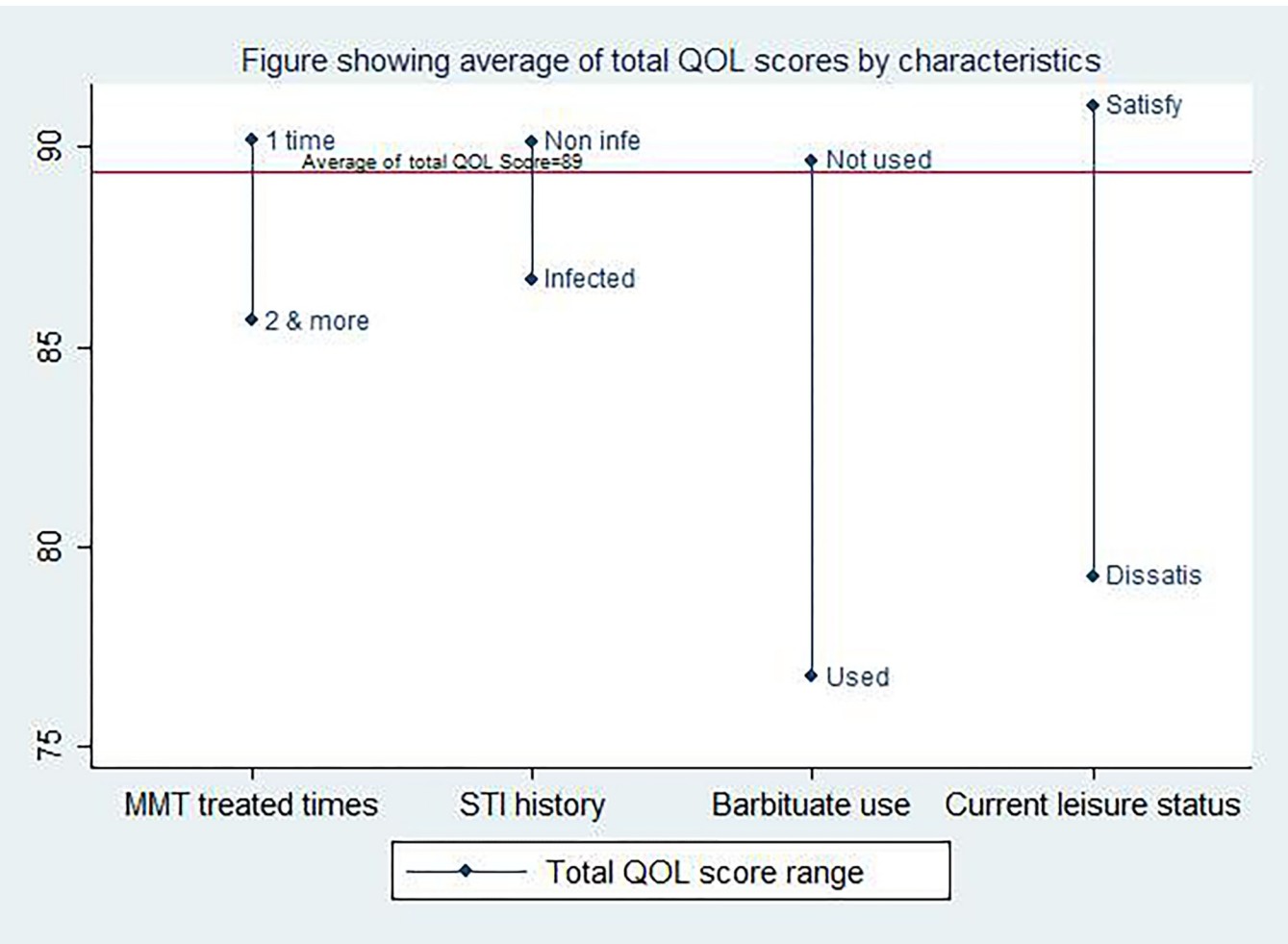

**Fig 1. Graph showing average of total quality of life (QOL) score differences by characteristics.** Y-axis shows the average of the total QOL score. X-axis shows significant characteristics associated with average of the total QOL score.

last 30 days proved a significant association with the higher QOL and those who didn't take barbiturates compared to those who used drugs.

Among the patient characteristics, employment status was one of the important factors that significantly differed the QOL total score (p = 0.0006) and all domains of life quality. This influential parameter was also consistent with recent studies which showed employment effects on increased QOL [21,22]. The unemployment lowered physical, psychological, environmental and social health of WHO-QOL score [23]. Those clients receiving income from temporary jobs and other non-criminal sources were significantly associated with the worse health-related QOL (HRQOL) scores in all domains [19]. The respondent's current leisure status was also highly associated with all domains contributing to the total QOL domain (p<0.001). "Leisure and social participation" was also an influential factor in the Lancashire Quality of Life Profile, [24]. Social network, and the relationship to family and friends were significant predictors and high QOL leisure time was associated with high overall QOL [25].

There were some limitations in the study design with the lack of ideal randomisation of the study participants. It was impossible to prove the QOL differences by the timeline or causal interventions of the methadone treatment from the cross-sectional study. With the limitation

of this study context, there were a few biases expected in this study. Selection bias from the specified selection criteria, observation bias as the Hawthorne Effect introduced by the service delivery organisations in conducting the survey, and recall bias from the respondents in answering the questions related to feeling and action during the specified timeline in the questions were observed. Due to the inclusion of the current methadone clients who were satisfied with the current treatment, there could be limitations in reflecting of methadone clients who were dropped out of the programme due to their poor quality of life or dissatisfaction with available methadone services.

In conclusion, methadone maintenance therapy is beneficial for patients who wish to improve their quality of life if prescribed effectively to the individual need of the patients. With over 80mg of methadone dose, first-time enrolled patients had higher quality of life than frequently enrolled patients. It seemed to prove that patients were retained in the programme and their quality of life was better than others if the dose was higher with individual necessity. In addition, satisfaction with the current leisure status of the patient is vital to be considered for maintaining the patient quality of life. Interventions addressing these factors associated with QOL will improve the consistent service utilisation and the quality of life among methadone patients. In addition, periodic research addressing the benefit of continuation of the methadone treatments is recommended to improve the quality of life of the methadone patients and program improvement.

## Supporting information

**S1 Text. Inclusivity in global research.**
(DOCX)

## Acknowledgments

The authors would like to thank all survey respondents and Dr. Nanda Myo Aung Wan, DDTRU Programme Manager, in Myanmar who supported proposal development. Further appreciation goes to Dr. Ohnmar Thaung, U Thet Swe, Dr. Phyo Myat, Dr. Nay Lin, Dr. Myo Minn Minn and harm reduction organisations in Myanmar (Myanmar Anti-Narcotic Association, Burnet Institute, Asian Harm Reduction Network) for supporting survey data collection.

## Author Contributions

**Conceptualization:** Sun Tun, Vicknasingam Balasingam, Darshan Singh Singh.

**Data curation:** Sun Tun.

**Formal analysis:** Sun Tun, Vicknasingam Balasingam, Darshan Singh Singh.

**Funding acquisition:** Sun Tun.

**Investigation:** Sun Tun.

**Methodology:** Sun Tun, Vicknasingam Balasingam.

**Project administration:** Sun Tun.

**Resources:** Sun Tun.

**Software:** Sun Tun.

**Supervision:** Sun Tun.

**Validation:** Sun Tun.

**Visualization:** Sun Tun.

**Writing – original draft:** Sun Tun, Vicknasingam Balasingam, Darshan Singh Singh.

**Writing – review & editing:** Sun Tun.

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
