## [Decision Letter · Decision Letter 0]

7 Mar 2022

PGPH-D-21-01098

Factors affecting the quality of life (QOL) scores among methadone patients in Myanmar

Dear Dr. Tun,

Thank you for submitting your manuscript to PLOS Global Public Health. After careful consideration, we feel that it has merit but does not fully meet PLOS Global Public Health’s publication criteria as it currently stands. Therefore, we invite you to submit a revised version of the manuscript that addresses the points raised during the review process.

EDITOR'S COMMENTS: 

The reviewers have assessed the strengths and weaknesses of the manuscript and made recommendation of a revision. Kindly consider the recommendations for revision they have made.The title require minor modification to align with the study design. Suggested modification is “Factors associated with quality-of-life scores among methadone patients in Myanmar."The study has not sufficiently operationalized quality of life measures. This requires theoretical and conceptual clarification in the introduction section. Data collection and measures should be separated. Instrument development and measures should be a sub-heading describing what constitutes the variables in the instrument being measured and how these are linked with the explanatory variables and the outcome variable of quality of life.A statement about construct, content and item validity is required. Similarly, a statement of assurance of reliability or standardization of the instrument is equally essential.Data collection procedure should follow describing how participants provided their responses, including urine samples.What was measured for the stated domains? There is need to explain the scale applied and to describe how these were developed. Scaling and scoring of constructs including number of items operationalizing each domain,The statistical analysis appear casual for such a study demonstrating rigour of design and elucidation of dynamics. Measures using the Likert response scale without adjusting for measurement error inherent in transforming  responses from frequency distribution to scale development scores to give summaries of descriptive statistics of means and standard deviation has serious consequences of accuracy.The result under the WHO QOL scores of methadone patients describing quality of life scores showed serious flaw in the data analysis from the 5-response Likert scale demonstrating error in the use of a 1 to 5 scoring modality instead of zeroing the scale.The sub-section of “Factors associated with quality of life (QOL) total score” has serious issues with weak statistical analysis expressing findings from the study data. What is communicated is inappropriate for expressing weighted-aggregate scores for quality of life measure

We look forward to receiving your revised manuscript.

Kind regards,

Nnodimele Onuigbo Atulomah, PhD

Academic Editor

Journal Requirements:

1. Please include a complete copy of PLOS’ questionnaire on inclusivity in global research in your revised manuscript. Our policy for research in this area aims to improve transparency in the reporting of research performed outside of researchers’ own country or community. The policy applies to researchers who have travelled to a different country to conduct research, research with Indigenous populations or their lands, and research on cultural artefacts. The questionnaire can also be requested at the journal’s discretion for any other submissions, even if these conditions are not met.  Please find more information on the policy and a link to download a blank copy of the questionnaire here: https://journals.plos.org/globalpublichealth/s/best-practices-in-research-reporting#loc-inclusivity-in-global-research. Please upload a completed version of your questionnaire as Supporting Information when you resubmit your manuscript.

2. Please update the completed 'Competing Interests' statement, including any COIs declared by your co-authors. If you have no competing interests to declare, please state "The authors have declared that no competing interests exist"

3. We see that your study includes live participants, but you have not included an Ethics Statement. Please update your manuscript file to include an Ethics Statement subsection to your Materials and Methods section. It should include:

iii) (for human participants or donors) - A statement that formal consent was obtained (must state whether verbal/written) OR the reason consent was not obtained (e.g. anonymity)

4. In the online submission form, you indicated that "The [.dta] data used to support the findings of this study are available from the corresponding author upon the approval of the Centre for Drug Research.". All PLOS journals now require all data underlying the findings described in their manuscript to be freely available to other researchers, either 1. In a public repository, 2. Within the manuscript itself, or 3. Uploaded as supplementary information.

Additional Editor Comments (if provided):

Two reviewers assessed the quality of the study reported in this manuscript submitted for possible publication which sort to explore Factors associated with quality-of-life scores among methadone patients in Myanmar and to elucidate the dynamics involved in the problem phenomenon related to treatment modality of those recovering from the use of potentially deadly chemical substances. The reviewers felt that revision of observed weaknesses in the manuscript is corrected would make it acceptable for possible publication. I strongly recommend that the authors attend to the areas pointed out as requiring modifications in the manuscript.

Reviewers' comments:

Reviewer's Responses to Questions

**Comments to the Author**

1. Does this manuscript meet PLOS Global Public Health’s publication criteria? Is the manuscript technically sound, and do the data support the conclusions? The manuscript must describe methodologically and ethically rigorous research with conclusions that are appropriately drawn based on the data presented.

Reviewer #1: Yes

Reviewer #2: Partly

2. Has the statistical analysis been performed appropriately and rigorously?

Reviewer #1: Yes

Reviewer #2: Yes

3. Have the authors made all data underlying the findings in their manuscript fully available (please refer to the Data Availability Statement at the start of the manuscript PDF file)?

Reviewer #1: Yes

Reviewer #2: No

4. Is the manuscript presented in an intelligible fashion and written in standard English?

Reviewer #1: Yes

Reviewer #2: No

5. Review Comments to the Author

Reviewer #1: Reviewers Comment:

PGPH-D-21-01098

Factors affecting the quality of life (QOL) scores among methadone patients in Myanmar

Overall, the authors have done well. However, these issues should be addressed to enhance the quality of the stiudy:

1. The concluding part of the abstract mainly highlighted the aim of the study. It should point out the factor that affected the quality of life in the study. Lines 25 and 25

2. The objective of the study ahoul end the introduction. Line 60

3. The result was poorly discussed. It should be free from the results there in. Instead, the results need to be interpreted considering the result here. There should be comparison, contrast, and posible description of the meaning. Summarize key results with reference to the study objectives

4. How were the missing data addressed?

5. Clearly identify and define all the outcomes for the study

6. The paragraphs of the discussion section are rather too small and should be restructured

7. The eligibility criteria should be clearly defined and unambiguous. Apart from patients above 18 years old who gave their informed consent to participate in the study, what other factors was considered in the sellection?

8. What effort was made by the authors to address potential sopurces of bias in the study?

9. The study limitations should be stated in line with sources of potential bias

10. Most of the references are either incomplete or wrongly written. Some internet references do not have web address while those with web address do not have retrieval dates. All these should be tidied up.

11. Correct all typographical and sentence errors.

Thank you.

Reviewer #2: Overall, the topic is an interesting one and very informative .It is applicable to developed and developing countries alike. The authors however need to read through the manuscript and ensure that there is logical flow in the construction of sentences. The introduction in the abstract does not align with the objectives of the study and this should be corrected. The methodology needs to be revisited including clear description of how individual respondents were selected, details or reference to a protocol for urine testing for ilicit drugs, ethical considerations etc.

Other comments to the author are presented in the attached word document .

6. PLOS authors have the option to publish the peer review history of their article (what does this mean?). If published, this will include your full peer review and any attached files.

**Do you want your identity to be public for this peer review?** For information about this choice, including consent withdrawal, please see our Privacy Policy.

Reviewer #1: **Yes: **Ogbonna Brian Onyebuchi

Reviewer #2: No

---

## [Decision Letter · Decision Letter 1]

1 Aug 2022

Factors associated with quality of life (QOL) scores among methadone patients in Myanmar

PGPH-D-21-01098R1

Dear Dr. Tun,

We are pleased to inform you that your manuscript 'Factors associated with quality of life (QOL) scores among methadone patients in Myanmar' has been provisionally accepted for publication in PLOS Global Public Health.

Best regards,

Joseph El-Khoury, MD MSc FRCPsych

Academic Editor

Reviewer Comments (if any, and for reference):

Reviewer's Responses to Questions

**Comments to the Author**

1. If the authors have adequately addressed your comments raised in a previous round of review and you feel that this manuscript is now acceptable for publication, you may indicate that here to bypass the “Comments to the Author” section, enter your conflict of interest statement in the “Confidential to Editor” section, and submit your "Accept" recommendation.

Reviewer #1: All comments have been addressed

2. Does this manuscript meet PLOS Global Public Health’s publication criteria? Is the manuscript technically sound, and do the data support the conclusions? The manuscript must describe methodologically and ethically rigorous research with conclusions that are appropriately drawn based on the data presented.

Reviewer #1: Yes

3. Has the statistical analysis been performed appropriately and rigorously?

Reviewer #1: Yes

4. Have the authors made all data underlying the findings in their manuscript fully available (please refer to the Data Availability Statement at the start of the manuscript PDF file)?

Reviewer #1: Yes

5. Is the manuscript presented in an intelligible fashion and written in standard English?

Reviewer #1: Yes

6. Review Comments to the Author

Reviewer #1: None. The issues raised have been addressed.

7. PLOS authors have the option to publish the peer review history of their article (what does this mean?). If published, this will include your full peer review and any attached files.

**Do you want your identity to be public for this peer review?** For information about this choice, including consent withdrawal, please see our Privacy Policy.

Reviewer #1: **Yes: **Ogbonna Brian Onyebuchi
